# Programmable responsive hydrogels inspired by classical conditioning algorithm

Hang Zhang [1,3], Hao Zeng [2,3], Arri Priimagi[2] & Olli Ikkala [1]

Living systems have inspired research on non-biological dynamic materials and systems chemistry to mimic specific complex biological functions. Upon pursuing ever more complex life-inspired non-biological systems, mimicking even the most elementary aspects of learning is a grand challenge. We demonstrate a programmable hydrogel-based model system, whose behaviour is inspired by associative learning, i.e., conditioning, which is among the simplest forms of learning. Algorithmically, associative learning minimally requires responsivity to two different stimuli and a memory element. Herein, nanoparticles form the memory element, where a photoacid-driven pH-change leads to their chain-like assembly with a modified spectral behaviour. On associating selected light irradiation with heating, the gel starts to melt upon the irradiation, originally a neutral stimulus. A logic diagram describes such an evolution of the material response. Coupled chemical reactions drive the system out-of-equilibrium, allowing forgetting and memory recovery. The findings encourage to search non-biological materials towards associative and dynamic properties.

[1] Department of Applied Physics, Aalto University, P.O. Box 15100, FI 02150 Espoo, Finland. [2] Smart Photonic Materials, Faculty of Engineering and Natural Sciences, Tampere University, P.O. Box 541, FI-33101 Tampere, Finland. [3]These authors contributed equally: Hang Zhang, Hao Zeng. Correspondence and requests for materials should be addressed to A.P. (email: arri.priimagi@tuni.fi) or to O.I. (email: olli.ikkala@aalto.fi)

Biological systems have inspired biomimetic materials with fascinating properties, e.g., toughness, structural colours, catalytic activity, and superhydrophobicity[1,2]. In future, bioinspired materials are foreseen to mimic ever more complex functional and adaptive properties of dynamic biological systems. Various model systems for dissipative out-of-equilibrium self-assemblies have already been presented[3–6]. To allow a major progress towards life-like materials, the importance of systems chemistry has been emphasized for the structural and temporal programming of the functionalities by coupled authchemical reactions under out-of-equilibrium conditions[7–11].

One of the most relevant biological functions deals with the concept of learning[12]. Upon learning, a biological system adopts new behaviours or functions upon responding to external stimuli. In its full biological form, it involves formidable complexity. However, systematic approaches have shed light on its simplest forms in biological systems, i.e., habituation, sensitization, and classical conditioning[13]. Adopting a still more reductionist approach, a generic question could be posed: can even inanimate materials be designed to show programmed responses inspired by some algorithmic forms of learning, in resemblance to systems based on synaptic electronics or biochemical circuits showing associatively responsive behaviours[14–19]?

To address the above question, we explore whether artificial non-biological materials could be designed to show algorithmic responses inspired by associative learning[20,21]. Associative learning, i.e. conditioning, is a learning process where two or more distinct stimuli are associated with each other[19]. In Pavlov's seminal experiment, an unconditioned dog salivates (unconditioned response, UR) upon seeing food (unconditioned stimulus, US), while ringing a bell (neutral stimulus, NS) does not lead to salivation. However, upon conditioning by simultaneously ringing the bell and showing the food, a conditioned response (CR) to the neutral signal is associatively learned, after which salivation can be triggered also by the bell. The process involves the association of two independent stimuli stored in the memory, so that the original response can be triggered by a new stimulus. In this case, the stimuli are fully orthogonal and undergo independent pathways (visual or auditory sense), finally being associated in the brain that provides the memory. For artificial materials, it is very challenging to construct two entirely orthogonal pathways for a specific stimuli–response, due to the lack of a central neural network. However, in a reductionistic approach, a synthetic material may algorithmically show some aspects of associative learning, provided that the system is capable of responding to two different stimuli, combined with a built-in memory element.

Hydrogels have been shown to be relevant model systems for out-of-equilibrium self-assemblies and systems chemistry[5,9]. Here we introduce a programmable soft hydrogel-based model system, the responses of which are inspired by classical conditioning, even if being mechanistically different. The aimed response is the gel–sol transition, which can be triggered naturally by heating. We denote this stimulus as the unconditioned stimulus, adapting the denotation from the Pavlov's dog experiment[20]. Light irradiation with a selected spectral composition is an originally neutral stimulus, not leading to gel melting, i.e., response. The system is designed such that associating such an irradiation (the neutral stimulus) and heating (the unconditioned stimulus) triggers a memory, whereafter a subsequent sole irradiation with the given spectral composition allows the gel melting (algorithmically inspired by the conditioning process in dogs). The memory is constructed by gold nanoparticles that are initially individually dispersed in the hydrogel, which become assembled into linear chains upon stimuli association, which in turn modifies their optical absorbance. The memory of the gel can be time-wise programmed using a systems chemistry approach, which drives the hydrogel out-of-equilibrium and mimics forgetting. Out-of-equilibrium conditions also allow spontaneous recovery of the memory after extinction. Although the two stimuli are not fully orthogonal and the response pathways (heating and light-induced heating) are interdependent, which deviates from the ideal concept, the programmed hydrogel system algorithmically shows many aspects of biological learning, and thus encourages research in synthetic materials towards more associative and dynamic functionalities.

## Results

**Programming a hydrogel inspired by associative learning**. The hydrogel consists of pH-sensitive gold nanoparticles (AuNPs) embedded in a soft hydrogel network of agarose and a merocyanine-based photoacid (Fig. 1a). The photoacid allows reversible photo-switching of the pH between 5.4 and 3.8 in the aqueous solution (0.2 mM) upon irradiation in the visible range at the wavelength 455 nm (Supplementary Fig. 1)[22,23]. The AuNPs are modified by lipoic acid, chosen to respond to pH changes caused by the photoacid[24]. The composition of the system, such as the size of AuNPs and the gel concentration, has been optimized for fast response under mild conditions (Supplementary Figs. 2–6). The programmed response of the hydrogel is shown in Fig. 1b–e. Heating (Stimulus 1, S1) above $T_m \sim 33$ °C induces gel melting as the intrinsic response (Fig. 1b, Supplementary Fig. 7), whereas the AuNPs remain well dispersed as confirmed by TEM imaging and UV–Vis measurement. A clear plasmonic band can be seen at 520 nm before and after heating (Fig. 1b, right panel). This is due to sufficient electrostatic stabilization of the carboxyl groups at pH above 5, since heating does not affect the pH of the gel significantly (Supplementary Fig. 1). As the second stimulus (S2) we used laser irradiation at 635 nm (140 mW cm$^{-2}$), mixed with 455 nm (25 mW cm$^{-2}$). The photoacid absorbs strongly in the range between 380 and 460 nm, so that efficient photoacid activation can be achieved at this wavelength. The intensity at 455 nm is chosen to be just sufficient for the activation of photoacid, in order to avoid significant contribution to the photothermal heating inside the gel. Under irradiation (S2), the original gel does not melt due to the low absorption at 635 nm and thus insufficient heating (Fig. 1c, Supplementary Fig. 8). Note that in this case the interparticle electrostatic repulsion is in fact also reduced upon the pH decrease, but the plasmonic band remains at 520 nm, suggesting dispersed particles due to the stabilizing gel matrix. The presence of the gel network hinders the diffusion of the AuNPs, and thus no significant self-assembly into chains takes place even when the photoacid is activated.

The crucial step to achieve evolution in the response of the material is the self-assembly of the AuNPs by simultaneous exposure to light (635 + 455 nm) and heat (Fig. 1d). Once the gel melts upon heating, the AuNPs regain mobility and self-assemble into linear aggregates triggered by the irradiation-induced pH change (Supplementary Figs. 9 and 10). The formation of linear assemblies of AuNPs is dependent on different parameters such as ligand composition, pH, solvent, or salt[25]. In our system, the use of lipoic acid is important to achieve the linear self-assembly. It has been proposed that anisotropic electrostatic repulsion of AuNP dimers formed in the initial aggregation stage accounts for the rather linear configuration of the aggregates[26]. In the gel, the linear self-assemblies of AuNPs remain stable even after the pH recovery (light off) and re-gelation of the agarose (Fig. 1d), which we attribute mainly to interparticle hydrogen bonding and/or van der Waals attraction[27]. The self-assembled AuNPs can be only separated by increasing the pH of the solution to above 8, as we show in the later sections. The resulting spectral change of self-

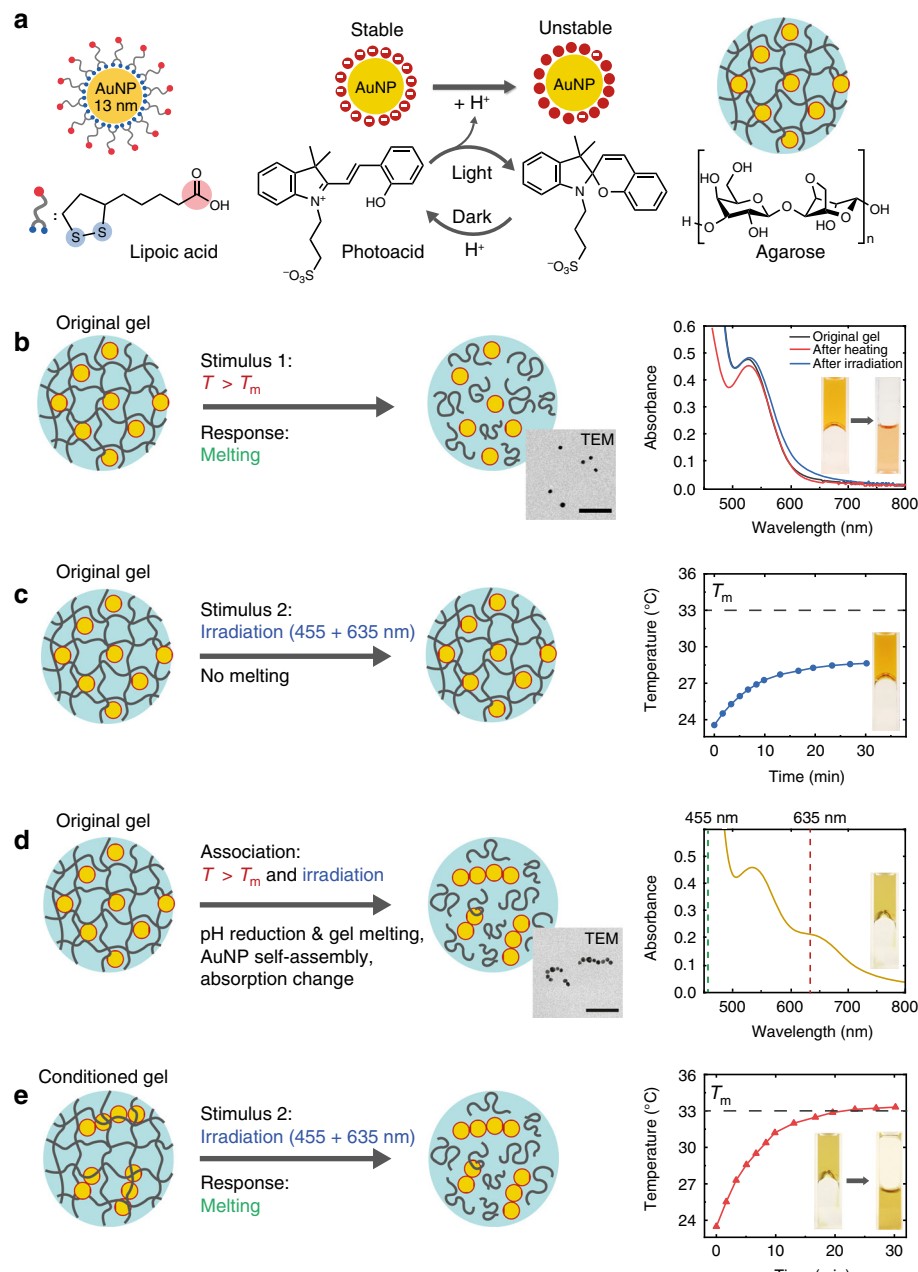

**Fig. 1** Programming hydrogel behaviour inspired by associative learning. **a** Lipoic acid-modified gold nanoparticles (AuNPs), photoacid, and agarose hydrogel. **b** Melting of the original gel upon heating above $T_m$. Inset: Transmission electron microscopy (TEM) image of initially dispersed AuNPs in the gel. Right panel: UV–Vis spectra of the original gel after irradiation or heating and photographs illustrating the gel melting by cuvette inversion. **c** Original gel after irradiation at $635 + 455$ nm. Right panel: the minor temperature increase and photograph of the gel upon irradiation. **d** Association of heating and irradiation. Inset: TEM image of AuNPs self-assembled in chains. Right panel: UV–Vis spectrum featuring enhanced absorbance at 635 nm and photograph of the programmed sample after re-gelation. The wavelength of 455 nm is marked as the second component of irradiation. **e** Melting of the conditioned gel upon irradiation at $635 + 455$ nm. Right panel: temperature increase and photographs of the programmed gel upon irradiation. Scale bar for TEM images: 100 nm

assembly is the appearance of a new plasmonic band around 635 nm due to the coupling of the AuNPs in the linear self-assemblies[28]. Note the subtlety that non-specific random AuNP aggregation would not result in a defined new band at longer wavelengths. This spectral change serves as the memory, which enables significantly enhanced photothermal heating at 635 nm due to the thermoplasmonic properties of AuNPs[29,30], and the gel thus melts upon irradiation (Fig. 1d, e). Hence the system has evolved to a new state, where upon association by two stimuli the AuNPs are self-assembled from individual particles into chains,

and the gel melting occurs upon the newly learned stimulus, i.e. light irradiation with the given wavelengths.

In order to allow generalization of the present programming concept inspired by the classical conditioning, Fig. 2 suggests the underlying logic diagram. In order to evolve to provide the given response based on a new stimulus, the material must possess a memory module that can be triggered by the external stimuli, and a read-out mechanism that modifies the behaviour upon switching of the memory[16]. In the gel, the memory is the spectral change due to the self-assembly of AuNPs to the linear chains

**Fig. 2** Logic circuits of the associative responses in the hydrogel. Note that the OR gate is a pure logical operator to represent a sustained memory once it has been switched on, without the need of further input from the learning AND gate. The memory of the hydrogel does not possess a real OR gate. The logic chart allows to search other materials with different stimuli and responses that show learning-inspired behaviour

that can be switched ON by the learning AND gate. This AND gate is achieved by the incorporation of the AuNPs/photoacid pair into the gel network, so that the self-assembly of the AuNPs is only possible when light (in particular the 455 nm component) and heat are both present. This ensures that the change of the material state is exclusively based on the association of two stimuli, i.e., heating-based gel melting, thus allowing the diffusion of AuNPs, and irradiation-based pH reduction, thus leading to the AuNP chain formation. The OR gate is necessary to sustain the memory once it has been switched on, which is accomplished by the stability of the linear self-assembly of AuNPs in the gel. On the other hand, the recalling AND gate is achieved by the photothermal effect due to the coupled surface plasmon resonance of the AuNPs, through which the material is able to respond to irradiation (in particular the 635 nm component).

That the gels can be constructed to obey the logic diagram shown in Fig. 2 is relevant to illustrate that non-biological systems can become programmable to allow responsive behaviour, inspired by associative learning. It is particularly important to note that the functionality described by the logic diagram is achieved without using electric circuits or biochemical networks[14–19]. However, there are still differences in comparison to the associative learning/conditioning processes in biological systems. On the one hand, in the case of the classical conditioning experiment of a dog[20], the stimuli (food/sound) are completely orthogonal: the two stimuli are processed through independent pathways: food-to-eyes-to-neural network and bell-to-ears-to-neural network. Each pathway undergoes physiological processes in mechanistically different ways, finally being associated in the memory (to salivate) after conditioning. In the gel system, the two stimuli (light and heat) are not ideally orthogonal. Specifically, light produces heat in the gel via photothermal effect and thus triggers the response. The intrinsic minor heating effect (given by 635 + 455 nm) exists even in a non-programmed sample because of the non-zero absorption at those wavelengths. Such intrinsic heating remains as a part of contribution to light actuation in subsequent programmed sample, while the newly obtained response (absorption from self-assembly of the AuNPs after stimuli association) contributes the major part of the light-induced heating (Supplementary Fig. 8). On the other hand, in our artificial system the light contains two wavelengths, as the 455 nm wavelength is mainly for the activation of the AuNP-based memory to allow its new response at the second wavelength at 635 nm, which is mainly responsible for the photothermal heating after conditioning. Although mechanistic deviations exist between natural and artificial systems, importantly, we can optimize the system so that its apparent behaviour mimics the association/conditioning algorithm (according to Fig. 2). Such an optimization includes selecting the working temperature, irradiation intensity, and spectra of the irradiation in a relatively flexible

and adjustable range. For example, if the gel is positioned in a warmer environment such as 28 °C, the laser intensity can be reduced so that the photothermal effect before conditioning is small enough as not to induce the melting, but becomes strong enough to trigger response after conditioning. The optimization is justifiable, as even systems capable of associative learning using biological cascades need optimization[19]. The importance is that this system shows the possibility for artificial materials (not based on biochemical networks or electric circuits) to exhibit the associative learning (conditioning) algorithm.

Importantly, the gel mimicking associative learning can be clearly distinguished from conventional shape memory materials[31,32]. This can be illustrated using a two-state classic shape memory material. Therein, the first stimulus prepares the system in a temporary shape (a kinetically trapped state), and the second stimulus recovers the equilibrium permanent shape fixed by the fabrication process. The shape memory materials also possess a memory (the equilibrium permanent shape), but the memory itself does not change after fabrication and the material always shows the same response by returning to the permanent shape. Shape memory materials therefore do not really evolve or learn to respond to a new stimulus, while the present hydrogel can be programmed to respond to an initially neutral stimulus.

**Timing dependence of the association process.** In biological systems undergoing classical conditioning, the efficiency of learning is highly dependent on the timing between the applied stimuli. US and NS may take place simultaneously, the NS may precede the US, or the US may precede the NS, denoted as simultaneous, forward, or backward conditioning, respectively. Forward and simultaneous conditioning are the fastest, while backward conditioning is less effective or even inhibitory[33]. This is presumably because the NS no longer predicts the appearance of US in the case of backward conditioning, where the NS is applied after the US[33], so that such an association will not be beneficial to the organism. The association process in the gel shown in Fig. 3a–c is in line with such observations. When the irradiation precedes or coincides with the onset of heating, the learning efficiency, as manifested by the increase in absorbance at 635 nm of the gel, is comparable in forward and simultaneous association (Fig. 3a, b). In contrast, backward association is less effective, and the increase of absorbance is roughly 70% of that resulting from the simultaneous process (Fig. 3c). This can be attributed to the viscosity increase of the solution upon removal of the heating (Supplementary Fig. 11)[34], which slows down the diffusion and thus the self-assembly of the AuNPs. Consequently, after the backward association, the gel does not melt upon irradiation, since the temperature stays below the $T_m$ of the gel, though the photothermal effect is stronger compared to the

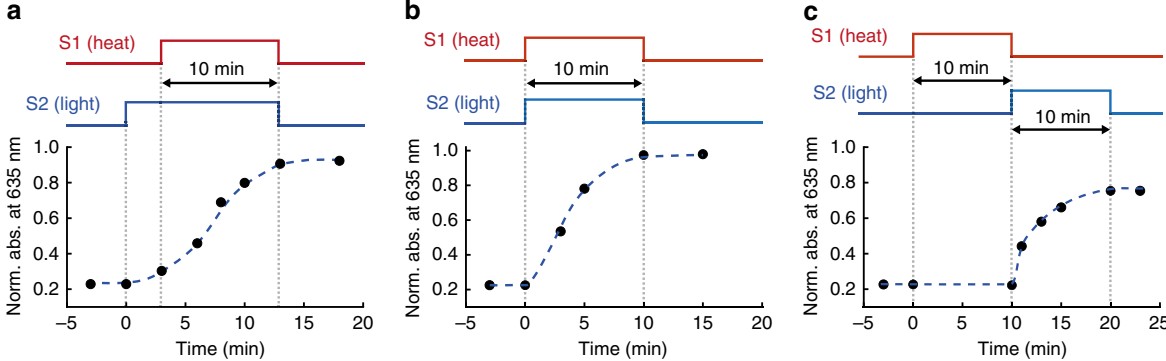

**Fig. 3** Timing dependence of the association process. Forward (**a**), simultaneous (**b**), and backward (**c**) association in the gel. The absorbance at 635 nm is normalized to the absorbance of the gel after simultaneous conditioning. Dashed lines are to guide the eye

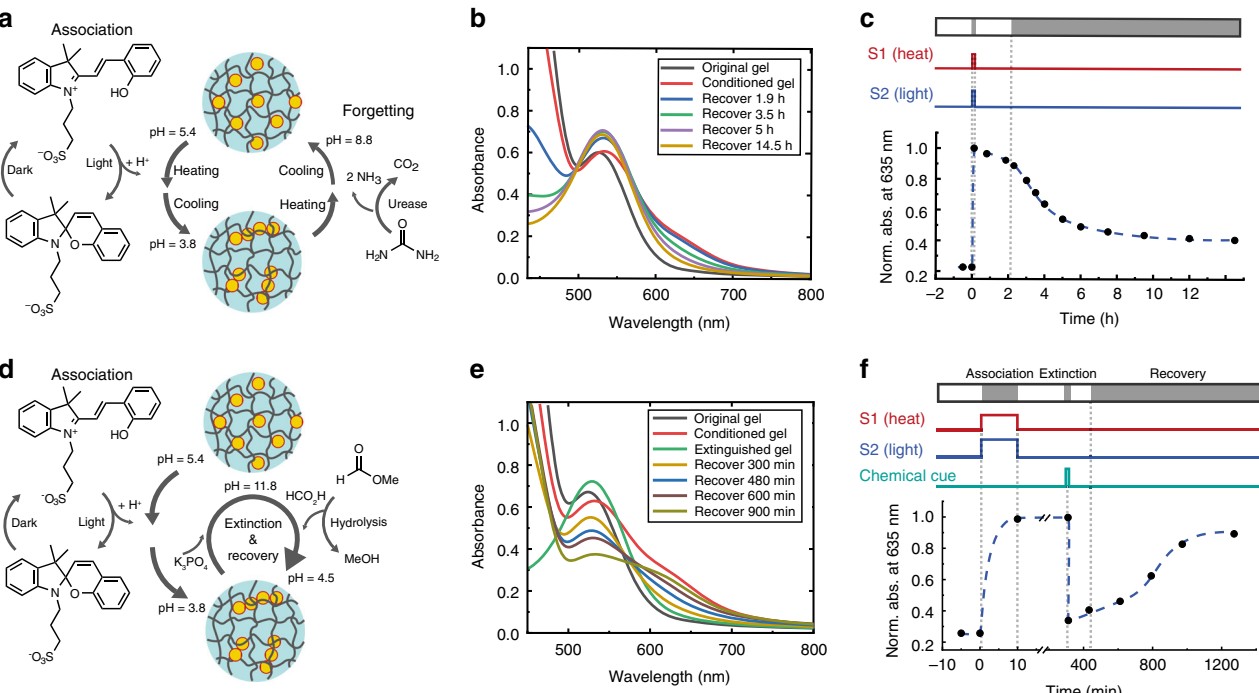

**Fig. 4** Temporally programming the hydrogel response using clock reactions. **a–c** Forgetting of the conditioned memory. **d–f** Extinction and spontaneous recovery of the memory. **a** Schematic illustration of coupled chemical reactions for the forgetting experiment. **b** The absorption spectra showing the acquisition and subsequent forgetting of the memory with time. **c** The normalized absorption at 635 nm as a function of time, showing the partial disassembly of the AuNPs due to increased pH. **d** Schematic illustration for the extinction and recovery of the memory. **e** The absorption spectra showing the acquisition and subsequent extinction and spontaneous recovery of the memory. **f** The normalized absorption at 635 nm as a function of time, showing the initial disassembly of the AuNPs upon chemical cue and subsequent recovery of the memory. Dashed lines are to guide the eye. Conditioning starts at $t = 0$ min

original gel (Supplementary Fig. 12). The ability to differentiate the temporal relationship between the unconditioned and neutral stimuli is intriguing and shows surprising similarity to the timing dependence of classical conditioning[33].

**Forgetting and spontaneous recovery of the memory.** Inspired by associative learning, the above results describe the gel behaviour in the equilibrium state, where the memory stays unchanged after the conditioning-inspired programming. As a step further, we expect that programming the time domain of the response and potentially driving the system out-of-equilibrium are needed to mimic more subtle aspects. The classical conditioning may involve several stages, such as acquisition, extinction, and spontaneous recovery[20,35]. Inspired by this, we show in

Fig. 4 the possibility of temporally programming the memory of the hydrogel using coupled chemical reactions, i.e., systems chemistry. In Fig. 4a–c, the original gel contains additional 20 mM of urea and 5 μg mL$^{-1}$ of urease as an internal clock to trigger the forgetting process. The urease catalyses the hydrolysis of urea, resulting in the production of 2 eq. of ammonia and 1 eq. of carbon dioxide. The temporal profile of the solution pH can thus be programmed depending on the ratio and concentration of the two components[36]. In this case, the memory of the gel is in fact modified to be the AuNP/agarose pair together with the urea/urease pair, where the latter enables metastable assembly of the AuNPs. In the gel, the pH remains almost unaffected by the presence of urea/urease on the time scale of ~10 min, which thus enables acquisition of the memory upon association between the two stimuli. Yet the programmed gel is not in the equilibrium

state. The forgetting process takes place as the result of urea hydrolysis, which slowly increases the pH to around 8.8 in 12 h. As the gelling point is below room temperature, the conditioned sol was left in the liquid state at room temperature to facilitate the disassembly of the AuNPs triggered by the pH change, resulting in the drop of absorbance (memory) as shown in Fig. 4b, c. The high pH required for the disassembly could be the result of interparticle van der Waals attraction/hydrogen bonding that has to be counter-balanced by strong electrostatic repulsion from the deprotonation of the carboxyl groups on the AuNP surfaces. The decrease of the absorbance at 635 nm is significant, indicating that the AuNPs are well protected by the ligands during the self-assembly process so that they can also be disassembled. In addition, the absorbance around 450 nm dropped as a result of the deprotonation of photoacid at high pH. The forgetting profile is reminiscent of the Ebbinghaus' forgetting curve: the memory decreases with time, yet a small portion of it is retained over an extended period[37]. As a result, the gel no longer melts upon irradiation (Supplementary Fig. 12), and can be considered as having forgotten the association of the two stimuli. To be noted is that the association and forgetting process only works once after preparation of the gel within a short time window (~ 10 min), in contrast to the flexible and rewritable memory in real biological systems.

The memory can also be extinguished by an external stimulus followed by a spontaneous recovery (Fig. 4d–f). The extinction is induced by a chemical cue containing a potassium phosphate buffer ($K_3PO_4$) and methyl formate. Even if mechanistically different, this is inspired by the extinction processes in biological systems, which are triggered by repeated exposure to the NS without US. After addition of the chemical cue to the conditioned gel, the pH of the melted gel increases instantaneously to 11.8 due to the buffer (20 mM), which leads to disassembly of the AuNPs and thus fast extinction of the memory. Subsequently, the spontaneous hydrolysis of the methyl formate (240 mM) results in the formation of formic acid and thus a controlled decrease of the pH to below 5.0 in 20 h[38]. As the pH decreases, self-assembly of the AuNPs again takes place spontaneously, where the gel was kept in liquid state to allow diffusion of the AuNPs. Since the gelling point of the 0.3% agarose gel is below room temperature, the gel just has to be left at room temperature after conditioning to allow the diffusion to take place. The absorbance at 635 nm therefore gradually recovers after the extinction (Fig. 4e). The decrease of the absorbance around 530 nm is due to the protonation of the photoacid during the recovery. This kinetically controlled pH change using phosphate buffer and methyl formate thus enables the extinction of the memory and the following spontaneous recovery. After extinction, the gel does not respond to irradiation, but subsequently regains the ability to melt upon irradiation once recovered (Supplementary Fig. 12).

The temporally programmed memory processes shown in Fig. 4 are essentially out-of-equilibrium, in contrast to the equilibrium "learning" demonstrated in Figs. 1–3, where the memory remains unchanged after programming. The acquired memory is only in a temporally stable state in Fig. 4a–c, which gradually decays with time due to the kinetically controlled chemical reaction. The system reaches equilibrium again only after the memory is almost lost (disassembly of the AuNPs). The same applies to the process in Fig. 4d–f, where the memory is temporarily extinguished, followed by a spontaneous recovery. Essentially, these are all chemical approaches inspired by the biological phenomena using clock reactions to program the lifetime of AuNP assemblies, which may be useful for out-of-equilibrium systems. These results demonstrate the possibility to manipulate the (primitive) memory of the gel system, inspired by processes that also operate under out-of-equilibrium conditions.

## Discussion

Associative responsive properties have been exploited in systems with reversible polymerization[18] and enzymatic reactions[19]. Very often, a modification of chemical composition is needed during the association process, for example by adding enzyme into reaction[19], similar to the extinction and spontaneous recovery experiments in our programmed hydrogel (Fig. 4d–f). Although these approaches provide valuable examples of artificial systems inspired by biological processes, we believe that keeping the chemical composition unchanged in the association process (Figs. 1, 3 and 4a–c) will bring the analogy closer towards Pavlov's classical conditioning experiments. Besides, such chemically fixed systems allow realizations of future devices with dynamic adaptation—once fabricated, the functions can be switched on via associative learning and recovered by forgetting—without the addition of any post-synthetic chemical fuels. Further clarifications of similarities and differences between programmed hydrogel and the reported enzyme-based systems are not enumerated here due to the complexity of each system. We refer readers to the Peer Review File in Supplementary Information for more details.

Summarizing, we have shown that an inanimate soft hydrogel can be designed to exhibit apparent behaviours resembling associative learning, which has been considered as one of the elementary forms of learning. This requires materials with at least two responses to different stimuli and one memory element. Therein, the material learns to respond to an initially neutral stimulus (selected spectral composition of light) through an associative process, during which the material is exposed to two stimuli (light and heat), as shown in the presented logic diagram. The memory of the system can be temporally programmed by driving the system out-of-equilibrium using clock reactions, which shows algorithmic resemblance to forgetting and recovery of memory in biological systems. The sol–gel transition in hydrogels with an artificial memory may find applications, e.g., in intelligent drug delivery or cell culture, and the concept may be extended to other material systems beyond gels following the demonstrated logic flow diagram, based on other functional groups and fields, e.g., magnetic fields. Admittedly, the content of learning in the demonstrated hydrogel/nanoparticle systems is prescribed to preselected stimuli when compared to the complex and adaptive behaviour in biological systems capable of responding to a wide variety of stimuli[39], and the underlying mechanism is distinct from the natural processes. Nevertheless, our system offers selectivity towards stimuli, and allows considerable flexibility for new properties and functions to be engineered (e.g., forgetting/recovery). The possibility to program a hydrogel to mimic classical conditioning-type response, as well as the temporal stability of the memory under out-of-equilibrium conditions, is conceptually intriguing. We envision that designing complex behaviours coupled with engineered physical properties of artificial materials may provide unforeseen routes and applications eventually mimicking living systems that can autonomously adapt.

## Methods

**Synthesis and modification of AuNPs.** To synthesize gold nanoparticles with an average diameter of 13 nm, the seeded growth method in the presence of tannic acid was used[40]. All solutions were freshly prepared in deionized water (18.2 MΩ; DirectQ 3 UV; Millipore) before the reaction. Briefly, 150 mL of 2.2 mM trisodium citrate dihydrate (99.5%; Sigma-Aldrich) was mixed with 0.1 mL of 2.5 mM tannic acid (ACS reagent; Sigma-Aldrich) and 1 mL of 150 mM potassium carbonate (99.995%; Sigma-Aldrich) in a 250 mL three-neck flask. The solution was heated to 70 °C in an oil bath. Subsequently, 1 mL of 25 mM HAuCl$_4$·3H$_2$O (>99.9%; Sigma-Aldrich) solution was quickly injected under vigorous stirring, and the solution was kept at 70 °C for another 5 min. In this way, gold seeds with an average diameter of 3.5 nm were prepared. Directly after the formation of gold seeds, 55 mL of this solution was extracted, and 55 mL of 2.2 mM citrate solution was added. After the

temperature of the solution reached again 70 °C, 0.5 mL of 25 mM HAuCl₄·3H₂O solution was injected to initiate the growth of the seed. After 10 min, an identical injection was carried out. This growth procedure including extraction, addition of citrate, and two injections of gold precursor was repeated for six times to obtain the 13 nm AuNPs. For the synthesis of AuNPs of other sizes, see Supplementary Information[41,42]. Characterization of the AuNPs was carried out by transmission electron microscope (Tecnai 12).

For modification of the AuNPs, lipoic acid (>99%; Sigma-Aldrich) was first dissolved in ethanol (99.5%; ALTIA Oyj) to make a 10 mM solution. The pH of 10 mL AuNP solution was adjusted to ~11 by adding 20 µL of 1 M NaOH (98%; Sigma-Aldrich) solution. The freshly prepared lipoic acid solution (1 mL) was added dropwise to the AuNP solution under stirring. The solution was then shaken overnight on an orbital shaker. Subsequently, the solution was centrifuged at 15 °C for three times (24,000g for 30 min; Avanti J-26XP centrifuge; Beckman Coulter), each time with removal of the supernatant after centrifugation, dilution with water to 20 mL, and addition of 20 µL of 1 M NaOH solution. After the last centrifugation, the precipitate was collected and diluted with water to a final volume of 1 mL. The modified AuNPs were always prepared freshly before use.

**Synthesis of photoacid.** Photoacid was synthesized according to literature[22,23]. First, 2,3,3-trimethylindolenine (3.18 g, 20 mmol, 98%; Sigma-Aldrich) was mixed with 1,3-propanesultone (2.44 g, 20 mmol, >99%; Sigma-Aldrich) and then heated to 90 °C for 6 h under nitrogen protection. The product was collected as a purple solid by filtration and washed thoroughly with diethyl ether. The dried product (600 mg, 2.14 mmol) and 292 mg of 2-hydroxybenzaldehyde (2.4 mmol, >99.0%; Sigma-Aldrich) were then added to 12.0 mL of anhydrous ethanol, and the solution was refluxed overnight. The resulting orange solid was collected by filtration and rinsed thoroughly with ethanol (5 × 10 mL). The final product was dried in vacuum and stored in a freezer (−20 °C) under nitrogen protection. The pH of the photoacid solution was measured by a Mettler Toledo SevenExcellence pH meter.

**Preparation and conditioning of the hydrogel.** An amount of 30 mg of agarose (ultra-low gelling temperature; A5030; Sigma-Aldrich) was dissolved in 10 mL of water in a pre-rinsed glass vial by heating to 70 °C and vortexing. In total, 0.78 mg of photoacid was added to this solution and sonicated at 35 °C until full dissolution. The resulting solution (1.92 mL) was mixed with a certain amount of 1 M NaCl solution (30 µL for the gels in Fig. 1 and 40 µL for the gels in Fig. 4 for faster self-assembly) in a disposable cuvette (BRAND; Sigma-Aldrich). Subsequently, 35 µL of the AuNPs solution was added and mixed thoroughly. The cuvette was sealed by Parafilm M and stored in fridge (4 °C) overnight for gelation. The reference gel was prepared in the same way except that water was added instead of AuNPs. For Fig. 4a–c, additional 1 M urea solution and 1 mg mL⁻¹ urease from Canavalia ensiformis (1 U mg⁻¹; Sigma-Aldrich) solution were added, so that the final concentration was 20 mM for urea and 5 µg mL⁻¹ for urease. This solution was conditioned directly after preparation. For Fig. 4d–f, a certain amount of 1 M K₃PO₄ solution and liquid methyl formate were injected after conditioning, so that the final concentration after mixing equalled 20 mM for the phosphate and 240 mM for methyl formate. The gel was kept at room temperature for the recovery process.

Association between the two stimuli was done in a water bath at 50 °C under irradiation that consists of a 635 nm laser (140 mW cm⁻²; Roithner Lasertechnik) and a 455 nm LED (25 mW cm⁻²; Thorlabs). Photothermal experiments were done in air and the temperature was monitored by an infrared camera (T62101; FLIR). Optical images were taken using a digital camera (5D Mark III, Canon). UV–Vis spectra were measured on an Agilent Cary 5000 UV-Vis spectrometer.

## Data availability
The data that support the findings of this study are available from the corresponding authors upon request.

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

## Acknowledgements

We thank Dr. Nonappa for the help with TEM, Dr. Emilie Ressouche for the help with photoacid synthesis, Lahja Martikainen for the help with rheological measurement of the gel, Anik Nath for the help with urea/urease experiments, and Prof. Jaakko Timonen, Prof. Francoise Winnik, Prof. Petri Ala-Laurila, and Prof. André Gröschel for comments on the manuscript. We acknowledge the provision of facilities and technical support by Aalto University at OtaNano—Nanomicroscopy Center (Aalto-NMC). This work is supported by ERC (Advanced Grant DRIVEN, Agreement No. 680083; Starting Grant PHOTOTUNE, Agreement No. 679646) and Academy of Finland (Center of Excellence HYBER and competitive funding to strengthen university research profiles No. 301820, and a postdoctoral grant No. 316416). This work is part of the Academy of Finland Flagship Programme "Photonics Research and Innovation (PREIN)", Decision Number 320165.

## Author contributions

O.I. conceived the project, and O.I. and A.P. supervised the project. H. Zhang designed and fabricated the hydrogel. H. Zeng and H. Zhang carried out the photothermal experiments on gels. All authors analysed the data and wrote the manuscript.

## Additional information

**Competing interests:** The authors declare no competing interests.

