## [Peer Review File · Nature Communications]

Reviewers' comments:

Reviewer #1 (Remarks to the Author):

The manuscript discusses the operation of kinetically coupled stimuli into logic networks to imitate the concept of Pavlovian learning. The concept of learning is definitely one of the distinguishing features of life and future materials will incorporate such features to be more life-like and adaptive in their properties.

While the setting promises a highly interesting subject, one has to be more critical about the realization and the interpretation of the system characteristics.

The Pavlov dog is an example of associative learning. Two triggers are given and processed through orthogonal signal uptake mechanism (bell/footsteps of assistant -> ear -> brain; placement of food -> eye -> brain). One of the effectors undergoes a different signal processing, or better it is linked to the other trigger (on a neural network level), at some point so that the neutral trigger (sound) ends up in being a decisive trigger. In the case of the dog, the dog associates the bell/footsteps of the assistant with the provision of food.

Here the authors do in fact NOT show associative learning, but apply an important and decisive oversimplification to their system. The authors use light with two wavelengths to trigger two orthogonal switches/effects in a material but consider this as a single stimulus and effector route ("irradiation"). However, we live in the times of precision engineered photoswitches, and this needs to be considered from a simple responsive material perspective, but also here, more importantly, from a conceptual material perspective.

Irradiation at 455 nm is used for the photoacid (trigger A), while 635 nm is used for the photothermal heating (trigger B). Normal heating (trigger C) is used as the control/unconditioned response. Although both LEDs are switched on simultaneously, the mechanistic details are profoundly different to the Pavlov experiment.

The authors claim that "irradiation" is the conditional trigger that allows to associate it with the heating trigger in a process called associative learning. However, this is wrong, the system in fact just memorizes a state ("kinetically trapped Au-NP chains") and then uses a NON-associated third pathway/trigger to activate the memory for a function.

In detail, trigger A (455 nm) and trigger C (heat) are used for conditioning (trapping of Au-NP chains); while trigger B (635 nm) is then used in the "conditioned" state to trigger the photothermal heating effect reminiscent of the initial trigger C (heat). Since trigger A has absolutely nothing to do in the "conditioned" state with trigger C or effect C, there is absolutely no associative learning (because trigger A has not changed how it is being processed to allow for associative learning). This is a decisive mechanistic oversimplification that contradicts the process of associative learning.

To some extent the authors also omit stating the important wavelength in Figure 1D. Why is the needed wavelength not given? It is not 635 nm (as shown on the other parts of the figure).

Is this limitation really critical? Yes, the authors wish to address a novel and fundamentally important aspect of materials science. Given the visibility of a Nature Communications article, it is important to assure that the claims of the manuscript, and in this case on associative learning, are really upheld in the data. Otherwise literature will be swept with similar systems that claim related learning effects, while this is in fact not the case as laid out above. This is also an important comment for the authors, because making such significant claims on associative learning in materials should also withstand a close inspection of critical expert readers. At the same time, non-experts may not be aware of these subtle differences and wrongly link simple programming of

a material response to learning.

Overall, this rational makes it clear that claims of Pavlovian learning are incorrect and correspond to an overinterpretation and oversimplification of the system complexity needed for associative learning. They need to be removed. The manuscript rather shows “teaching” to stay with the authors’ language, or quite simply programming of a material response.

As a side note, chemical evolution of the photoacid would most likely allow to shift its absorption spectrum to the range of the photothermal heating. While this would reduce the amount of trigger signals, one would still need to consider – on a conceptual level – that the trigger is processed through two independent and not interdependent effector pathways (acidification to trigger assembly vs. photothermal heating). This again still cannot be considered associative learning.

Overall, the manuscript definitely shows a programming of a memory function through kinetic trapping, and the incorporation of the clock reactions are positive and valuable additions to simulate “forgetting”. However, the programming of a response state is rather routine (even so not routinely done via post-synthetic procedures), and such clock reactions are known and do not add sufficient novelty to warrant publication in *Nature Communication*. It is very well suited for a soft matter/materials journal. For such a publication, Figure 2 would need to be divided on the irradiation part in two triggers, and Figure 3 should be taken out of the context of learning and put in the context of time-correlated programming. Still all very interesting.

The literature selection lacks a few key papers: For instance: (a) “Associative Learning in biochemical networks” *J. Theor. Biol.* 2007, 249, 58. (b) “A bioinspired associative memory system based on enzymatic cascades” *Chem. Commun.*, 2013, 49, 6962.

In summary, this is a solid manuscript with very good data and no objections to the results obtained. However, the interpretation is an oversimplification of the processes of learning and it would set an inappropriate example if it was published under the conceptual placement of associative learning.

Reviewer #2 (Remarks to the Author):

In this article, O. Ikkala, A. Priimagi, and co-workers report a ‘Pavlovian gel’, that can ‘learn’ to respond to a neutral stimulus after conditioning. The agarose gel, that contains a photoacid and gold nanoparticles (AuNP) modified with lipoic acid, could be melted by an unconditioned stimulus (heat), however when light (635 and 455 nm) was irradiated into the sample nothing happened (neutral stimulus). The material could then learn to respond to light, when both heat and light were applied to the gel, the photo acid was activated, releasing protons to the medium; the AuNPs then aggregate linearly, creating a plasmonic band at 635 nm. This conditioning renders the gel responsive to light, making it possible to melt it by shining a 635 nm laser. The gel is then also coupled to out-of-equilibrium chemical reactions, making it possible for the gel to ‘forget’ the conditioning (urease urea chemical cycle) and the ‘extinction then recovery’ of the learning (hydrolysis of methyl formate).

The paper is very well written and easy to follow, and the experiments performed support the authors' claims. The ideas are very original, and will be beneficial to the field and can inspire soft matter / supramolecular chemists to pursue more life-like materials. My only criticism is that the out-of-equilibrium experiments are carried out in the melted form, which is needed for the AuNPs and chemicals diffusion, but it is not practical. The authors should more clearly indicate the limitations of their current approach. With that being said, I think that this manuscript fits well into the journal and should be accepted.

Reviewer #3 (Remarks to the Author):

The authors of the manuscript under review demonstrate a hydrogel system that upon simultaneous or successive exposure to heating and optical irradiation (through a range of different kinetic pathways), the dependence of gel responses, in this case melting, to optical stimuli can be tailored to mimic purely thermal cues, be reset and be recovered. The mechanism behind this altered gel behavior is reported to be the pH-dependent self-assembly (or disassembly) of lipoic acid-modified gold nanoparticles (AuNPs) in the liquid state and the resulting new plasmonic band around 635nm. The authors establish an analogy between classical Pavlovian conditioning, extinction, recovery and pathway-dependent modified, erased, reemerging gel responses, respectively, to a carefully tuned optical stimulus. To this end, the authors dub their system "Pavlovian hydrogel." The motivation and experimental results in the manuscript are verbally and visually very clearly presented.

However, it is my strong opinion that the analogy between Pavlovian conditioning and the modified gel responses to specific kinetic pathways, as concisely illustrated by a logic circuit in Fig. 2, is controversial. Thus, the main motivation and moral of the study are not well posed, as I will summarize below. The most concrete manifestation of that is lack of some very essential control experiments, to be listed below. That said, I cannot recommend the publication of manuscript as is or with minor revisions. My suggestion is to tone the analogy down significantly and present the experimental system as a proof of concept that combines a careful design of in- and out-of-equilibrium processes in the realm of systems chemistry, while clearly distinguishing it from the existing literature of shape memory materials. This seems a daunting task, however if established, the resulting quality of the manuscript may be eligible for a second round of review.

Major remarks:

1- Gel programming with classical conditioning: In Pavlov's research on dog physiology, food as an unconditional stimulus and bell as a neutral stimulus are independent, or say the two stimuli "orthogonal" to each other. In the gel system however, heat and light are not orthogonal: (1) In Fig. 1C rightmost panel, starting the illumination of gel at a higher temperature, such as 29-30oC could give rise to gel melting without an emergence of a new plasmonic band. By sharp contrast, a bell would never cause salivation without prior conditioning. (2) Naturally, as reported in the Materials and Methods, the gel solution was prepared much above the melting point (at 70oC) and then stored in fridge (4oC) overnight for gelation. What if the sample was irradiated before gelation? It appears that the initial temperature $T=24\text{oC}$ in experiments is also a fitting parameter to substantiate the analogy. (1) and (2) are missing control experiments with somewhat clear outcomes that would not serve the analogy.

On line 43, the correct expression should state "... the association of the two independent stimuli..."

Similarly, what if the gel was heated from 24oC up by a laser at another wavelength between 550-600nm with the same or higher power density? Again, the so-called "unconditioned stimulus" would likely directly give rise to the "conditioned response" without a learning stage.

Now the authors may argue that their analogy holds within the specific range of parameters that they have carefully tuned, the response to irradiation may be perceived as binary (melting, no melting) within the specific parameter range. But the number of parameters and (in the context of information theory) the "sloppy" response of the system to each of them makes the generalness of the concept highly questionable.

2- Timing of conditioning: Apparently the duration of heat is another control parameter: what if heat was given for a much shorter time (enough that the unconditioned response appears), even to raise the temperature slightly above the melting temperature T_m , where the viscosity is still high to slow down the self-assembly of AuNPs? This further breaks the binary nature of Pavlovian conditioning and in turn the analogy: beyond T_m the rate of self-assembly is still temperature-dependent. When the heat is applied for a much longer time in Fig. 3C, then the viscosity will be low enough to mediate self-assembly. This is contrary to the neutral stimulus in fact becoming inhibitory in classical Pavlovian picture.

The explanation for the inefficiency for backward conditioning provided in lines 139-141 is hard to follow: how much does the viscosity of the 0.3% g/mL agarose solution increase upon removal of heat (hard to read the data in Ref.30 as there it is not presented in a logarithmic scale)? If the viscosity does not change by an order of magnitude, one cannot really claim an inefficiency due to the slow rate of self-assembly of AuNPs in the liquid state.

So, if self-assembly is inefficient, effects of continuous variables (temperature) are at play, making the mapping between a binary response to a set of continuous responses elusive.

3- Forgetting and spontaneous recovery of memory: I am not sure how the Pavlovian picture describes "forgetting," perhaps a reference would help. It must in principle manifest itself as a relaxation towards equilibrium from a non-equilibrium or metastable state. The author's initial design of system disagrees with this as the gel system is in equilibrium in the absence of additional chemical cues, unless self-assembled chains of AuNPs are metastable.

In any case, both forgetting and spontaneous recovery must be associated with no additional chemical cues, whereas the authors used urea and urease for forgetting, and K₃PO₄ and methyl formate for spontaneous recovery. The addition of chemical cues in fact redefines the neutral stimulus as, e.g. illumination+urea+urease. Can the authors program their system in the first place with these modified ND, and then show forgetting and spontaneous recovery? For instance, K₃PO₄ and methyl formate would likely always disallow the self-assembly of AuNPs and either initial programming or later recovery must be compromised.

4- Materials and Methods: Absorbance measurements and methods are not described, this must be added.

Minor remarks:

1- Lines 58-59: How fast is "fast?" What two time scales do the authors compare?

2- Line 170: Did the authors mean "540nm?"

3- Fig. 4E: The green and blue curves are very hard to distinguish in print.

Answers to the Referees' comments:

Reviewer #1

Our general response:

First, we appreciate the reviewer's conclusion that this is a solid manuscript with very good data and no objections to the results obtained.

We also gratefully thank the reviewer for the in-depth critical comments that greatly helped to improve the manuscript.

Before going into the detailed responses towards the posed questions and criticism, we thank the reviewer for pointing out two relevant articles (*J. Theor. Biol.* 2007, 249, 58 and *Chem. Commun.*, 2013, 49, 6962), which describe associative learning and associative memory in biochemical networks.

The first article ("Associative learning in biochemical networks", *J. Theor. Biol.* 2007, 249, 58) includes clear definitions on the concepts of "learning", "associative memory", and "associative learning". Relevant for the present manuscript, the article states in the abstract: "*Associative learning is a form of learning whereby a system "learns" to associate two stimuli with one another. **Associative learning, also known as conditioning**, is believed to be a powerful learning process at work in the brain (associative learning is essentially "learning by analogy").*" The logic of our manuscript is in agreement with that definition, while extending the concept, nontrivially, from biochemical networks to model examples of "synthetic materials" (not exploiting electric devices).

To adapt more on the above-mentioned prior-art literature in biological networks, in the revision we primarily denote the present process as **inspired** by "associative learning" or "classic conditioning". See also the new title "**Programmable responsive hydrogels inspired by classical conditioning algorithm**", considering that we program the **apparent behaviour** of the gel to algorithmically follow the classical conditioning framework, **fulfilling the logic diagram Figure 2**, even if following mechanistically different pathways. Additionally, the limitation in the interdependent pathways is also discussed in a new paragraph.

The latter article, as suggested by the reviewer ("A bioinspired associative memory system based on enzymatic cascades", *Chem. Comm.*, 2013, 49, 6962), indeed turns particularly relevant for the present manuscript for comparison. There, the application of the "correct" input signal activates the system to yield the output signal, while the "wrong" signal results in no chemical changes. A simultaneous application of the "correct" and "wrong" signals ("training" step) results in the memory effect after which the system reacts to the application of the "wrong" signal as it would react to the "correct" signal, by producing the output signal.

Related to this article, we do see a conceptual difference. In *Chem. Comm.*, 2013, 49, 6962 the chemical composition is modified by adding new enzymes (as "signals") to allow a new response. By contrast, in our case, our chemical composition remains fixed, and the stimuli are physical external conditions. We found it important to keep the chemical composition fixed in the learning process, in analogy with the Pavlov's classical conditioning experiment where the state of the dog is kept as constant as possible.

Reviewer remark:

The manuscript discusses the operation of kinetically coupled stimuli into logic networks to imitate the concept of Pavlovian learning. The concept of learning is definitely one of the distinguishing features

of life and future materials will incorporate such features to be more life-like and adaptive in their properties.

While the setting promises a highly interesting subject, one has to be more critical about the realization and the interpretation of the system characteristics.

The Pavlov dog is an example of associative learning. Two triggers are given and processed through orthogonal signal uptake mechanism (bell/footsteps of assistant -> ear -> brain; placement of food -> eye -> brain). One of the effectors undergoes a different signal processing, or better it is linked to the other trigger (on a neural network level), at some point so that the neutral trigger (sound) ends up in being a decisive trigger. In the case of the dog, the dog associates the bell/footsteps of the assistant with the provision of food.

Here the authors do in fact NOT show associative learning, but apply an important and decisive oversimplification to their system. The authors use light with two wavelengths to trigger two orthogonal switches/effects in a material but consider this as a single stimulus and effector route (“irradiation”). However, we live in the times of precision engineered photoswitches, and this needs to be considered from a simple responsive material perspective, but also here, more importantly, from a conceptual material perspective.

Irradiation at 455 nm is used for the photoacid (trigger A), while 635 nm is used for the photothermal heating (trigger B). Normal heating (trigger C) is used as the control/unconditioned response. Although both LEDs are switched on simultaneously, the mechanistic details are profoundly different to the Pavlov experiment.

The authors claim that “irradiation” is the conditional trigger that allows to associate it with the heating trigger in a process called associative learning. However, this is wrong, the system in fact just memorizes a state (“kinetically trapped Au-NP chains”) and then uses a NON-associated third pathway/trigger to activate the memory for a function.

In detail, trigger A (455 nm) and trigger C (heat) are used for conditioning (trapping of Au-NP chains); while trigger B (635 nm) is then used in the “conditioned” state to trigger the photothermal heating effect reminiscent of the initial trigger C (heat). Since trigger A has absolutely nothing to do in the “conditioned” state with trigger C or effect C, there is absolutely no associative learning (because trigger A has not changed how it is being processed to allow for associative learning). This is a decisive mechanistic oversimplification that contradicts the process of associative learning.

Our response:

We thank for these comments, which encouraged us to carefully reconsider the text and concept. We think that it is the essence of bio-inspiration to mimic some (even if limited) aspects of biological complex behavior in simplified ways, allowing rational design of functions, as the biological systems are invariably complex. In this context, we agree that our system deviates from the real biological case, but we would like to argue that this is a justifiable simplification/adaptation, especially when looking only at the apparent behaviour of the hydrogel (as manifest in the logic diagram Figure 2). Taking the hydrogel to the prescribed stimuli, one can indeed observe a conditioning behaviour, algorithmically the same as the classical conditioning on dogs. We would like to emphasize that we are only inspired by the classical conditioning algorithmically in our artificial materials, and NOT mechanistically as compared to the complicated biological examples, which we highlight repeatedly in the revised manuscript.

Simplification has been exploited in most bioinspired systems, and also in the article suggested by the reviewer (*Chem. Commun.*, 2013, 49, 6962), which, in fact also uses a combination of two enzymes for the neutral stimuli (termed “wrong stimulus” in the reference), combining their specific effects as “one” composite stimulus. The first enzyme “HK” activates the memory (production of glucose-6-phosphate) during conditioning, while the second one “G6PDH” triggers the response after conditioning. This is

coincidentally a very similar strategy compared our use of the 455 nm and 635 nm light. Optimization was needed both in our case and in the case of the reference, as we needed to balance the stimuli and responses to demonstrate the associative learning = conditioning algorithm and the related logic diagram Figure 2.

Conceptually, we claim that this is the first “synthetic” model material not based on biochemical networks or electric circuits that provide simple learning functionalities apparently following the algorithms of conditioning = associative learning.

We have removed possibly misleading too direct linking to “associative learning”, “Pavlovian gel”, “conditioning”, etc. in the description of gel programming, and all the experimental process. Instead, the focus is laid on bioinspiration and the system chemistry approach, in which the programming of the gel is “inspired” by associative learning process.

Reviewer remark:

To some extent the authors also omit stating the important wavelength in Figure 1D. Why is the needed wavelength not given? It is not 635 nm (as shown on the other parts of the figure).

Our response:

We thank the reviewer for pointing this out. The two wavelengths were clearly stated in the main text and in the experimental section. The 455 nm was not included in the figure to avoid complication (the figure was already crowded) and to highlight the “memory” and the trigger (635 nm light) to retrieve the memory.

As the reviewer pointed out that this might be misleading, we have revised the figure accordingly, as shown below. (Stimulus 2, and Fig. 2d UV-Vis spectra.)

Reviewer remark:

Is this limitation really critical? Yes, the authors wish to address a novel and fundamentally important aspect of materials science. Given the visibility of a *Nature Communications* article, it is important to assure that the claims of the manuscript, and in this case on associative learning, are really upheld in the data. Otherwise literature will be swept with similar systems that claim related learning effects, while this is in fact not the case as laid out above. This is also an important comment for the authors, because making such significant claims on associative learning in materials should also withstand a close inspection of critical expert readers. At the same time, non-experts may not be aware of these subtle differences and wrongly link simple programming of a material response to learning.

Our response:

We agree that the different notions related to associative learning, associative memory, and conditioning need to be explained for the non-expert audience. Therefore, in the revised manuscript we include definitions according the article suggested to by reviewer (“Associative learning in biochemical networks”, *J. Theor. Biol.* 2007, 249, 58) with a reference thereof.

A sentence has been added in page 2:

Associative learning, i.e. conditioning, is a learning process where two or more distinct stimuli are associated with each other¹⁹.

In the manuscript, we emphasize that we only program the apparent behaviour of the hydrogel to allow the logic diagram (Figure 2), which is inspired by the “associative learning”, not mechanistically following the learning complexity in biological examples. We have added the following paragraph to highlight the mechanistic difference between our gel system and the real conditioning example in biology:

The following text has been added in page 7:

That the gels can be constructed to obey the logic diagram shown in Fig. 2 is relevant to illustrate that nonbiological systems can become programmable to allow responsive behaviour, inspired by associative learning. It is particularly important to note that the functionality described by the logic diagram is achieved without using electric circuits or biochemical networks. However, there are still differences in comparison to the associative learning/conditioning processes in biological systems. On one hand, in the case of the classical conditioning experiment of a dog²⁰, the stimuli (food/sound) are completely orthogonal: the two stimuli are processed through independent pathways: food-to-eyes-to-neural network and bell-to-ears-to-neural network. Each pathway undergoes physiological processes in mechanistically different ways, and finally associates to the same memory (to salivate) in the brain after conditioning. In the gel system, the two stimuli (light and heat) are not ideally “orthogonal”. Specifically, light produces heat in the gel via photothermal effect and thus triggers the response. The heating effect (given by 635 + 455 nm) exists intrinsically even in a non-programmed sample because of the non-zero absorption at those wavelengths. Such intrinsic heat remains as a part of contribution to light actuation in subsequent programmed sample, while the newly obtained response (absorption from self-assembly of the AuNPs after stimuli association) contributes another part of the light induced heat (Supplementary Fig. 8). On the other hand, in our artificial system the light contains two wavelengths, as the 455 nm wavelength is mainly for the activation of the AuNP memory to allow its new response at the second wavelength at 635 nm, which is mainly responsible for the photothermal heating after conditioning. Although mechanistic deviations exist between natural and artificial systems, importantly we can optimize the system so that its apparent behaviour mimics the association/conditioning algorithm (according to Fig. 2). Such an optimization includes selecting the working temperature, irradiation intensity and spectra of the irradiation, in a relatively flexible and adjustable range. For example, if the gel is positioned in a warmer environment such as 28°C, the laser intensity can be reduced so that the photothermal effect before conditioning is small enough as not to induce the melting, but becomes strong enough to trigger response after conditioning. The optimization is justifiable, as even systems capable of associative learning using biological cascades need optimization¹⁹. The importance is that, this system shows the possibility for artificial materials (not based on biochemical networks or electric circuits) to exhibit the associative learning (= conditioning) algorithm.

Reviewer remark:

Overall, this rational makes it clear that claims of Pavlovian learning are incorrect and correspond to an overinterpretation and oversimplification of the system complexity needed for associative learning. They need to be removed. The manuscript rather shows “teaching” to stay with the authors’ language, or quite simply programming of a material response.

Our response:

We thank for the comment. As described above, we revised the denotation (throughout all the text) as “inspired by associative learning”, according to *Chem. Commun.*, 2013, 49, 6962 to tone down too direct linking to Pavlovian learning. We have also changed the caption of the logic flow Figure 2 into, “Logic circuits of the programmable responses in the present material system”.

Reviewer remark:

As a side note, chemical evolution of the photoacid would most likely allow to shift its absorption spectrum to the range of the photothermal heating. While this would reduce the amount of trigger signals, one would still need to consider – on a conceptual level – that the trigger is processed through two independent and not interdependent effector pathways (acidification to trigger assembly vs. photothermal heating). This again still cannot be considered associative learning.

Our response:

As the reviewer suggests, it is indeed possible to reduce the neutral stimulus to one wavelength of light by chemical modification of the photoacid or by using a different type of nanoparticles, such as silver that has plasmonic bands at shorter wavelengths. This would drive the system closer to the real

conditioning process and can be of interest for future research. Still, the logic diagram can be fulfilled, even with the present system.

On the other hand, the reviewer pointed out that the pathways are interdependent. However, this is another mechanistic detail of the biological associative learning and does not invalidate our claim for the algorithmic programming in the artificial system. In fact, in the reference suggested by the Reviewer “Associative Learning in biochemical networks, *J. Theor. Biol.* 2007, 249, 58.”, the proposed effector pathways for associative learning are also interdependent as we explicate below.

Basic reactions:

1. $A \xrightarrow{\text{Trigger 1}} A + A$ (Replication of species A triggered by 1)
2. $B \xrightarrow{\text{Trigger 2}} B + B$ (Replication of species B triggered by 2)
3. $A + B \xrightleftharpoons{\text{spontaneous}} (A - B)$ (Reversible polymerization of A and B)
4. $(A - B) \xrightarrow{\text{Trigger 1 or 2}} (A - B) + (A - B)$ (Replication of the polymer $(A - B)$)

Association (conditioning):

5. $A + B \xrightarrow{\text{Trigger 1 and 2}} A + A + B + B + (A - B)$ (Association of trigger 1 and 2 using $(A - B)$ as the memory)

Recalling:

6. $A + B + (A - B) \xrightarrow{\text{Trigger 1}} A + A + B + (A - B) + (A - B)$
7. $(A - B) \longleftrightarrow A + B$ (Trigger 1 eventually results in the replication of B)

However, B can also be produced by trigger 2 via the $(A - B)$ polymer due to the presence of A , through the following steps:

8. $A + B \xrightarrow{\text{Trigger 2}} A + B + B$ (Replication of B)
9. $A + B \longleftrightarrow (A - B)$ (Polymerization of A and B)
10. $(A - B) \xrightarrow{\text{Trigger 2}} (A - B) + (A - B)$ (Replication of $(A - B)$)
11. $(A - B) \longleftrightarrow A + B$ (Reverse reaction of polymerization)

In summary, Trigger 2 alone results in the production of B and $(A - B)$, and Trigger 1 results in the production of A and $(A - B)$ after conditioning. Both effectors trigger the production of $(A - B)$, based on which the association is established. Therefore, Triggers 1 and 2 are **NOT** fully orthogonal. Furthermore, these reaction networks do not perfectly fit into the classical conditioning framework, since the trigger 1 alone without conditioning could already produce B via the $(A - B)$ polymerization and replication route (similar to the reactions 8-11). Nevertheless, we agree that the association processes are established in this paper, and the use of “associative learning” in the sense of the algorithm is acceptable. Similarly, we would argue that our claim for algorithmic programming of the hydrogel response with classical conditioning is justified, despite the mechanistic difference in the choice of neutral stimuli and effector pathways. However, to avoid confusion we have significantly toned down the analogy between our system and associative learning throughout the entire text.

We also added a few sentences to discuss the limitation in the interdependent pathways.

Text has been added to page 7:

On one hand, in the case of the classical conditioning experiment of a dog²⁰, the stimuli (food/sound) are completely orthogonal: the two stimuli are processed through independent pathways: food-to-eyes-to-neural network and bell-to-ears-to-neural network. Each pathway undergoes physiological processes in mechanistically different ways, and finally associates to the same memory (to salivate) in the brain after conditioning. In the gel system, the two stimuli (light and heat) are not ideally “orthogonal”. Specifically, light produces heat in the gel via photothermal effect and thus triggers the response. The heating effect (given by 635 + 455 nm) exists intrinsically even in a non-programmed sample because

of the non-zero absorption at those wavelengths. Such intrinsic heat remains as a part of contribution to light actuation in subsequent programmed sample, while the newly obtained response (absorption from self-assembly of the AuNPs after stimuli association) contributes another part of the light induced heat (Supplementary Fig. 8).

Reviewer remark:

Overall, the manuscript definitely shows a programming of a memory function through kinetic trapping, and the incorporation of the clock reactions are positive and valuable additions to simulate “forgetting”. However, the programming of a response state is rather routine (even so not routinely done via post-synthetic procedures), and such clock reactions are known and do not add sufficient novelty to warrant publication in *Nature Communication*. It is very well suited for a soft matter/materials journal. For such a publication, Figure 2 would need to be divided on the irradiation part in two triggers, and Figure 3 should be taken out of the context of learning and put in the context of time-correlated programming. Still all very interesting.

Our response:

We thank the reviewer for the suggestions. The logic flow diagram has been modified to be more accurate in describing the stimuli and responses.

We still claim that this is the first artificial model material (not using electric devices) showing responses inspired by associative learning. Even more generally, it shows that synthetic material systems can be designed to fulfil complicated responses, explained by the logic diagrams. This suggests to search other polymer/soft matter systems that also could obey logic flow diagrams in different functions. Despite its limitations, we believe that our article drastically expands the view of soft matter science to search “synthetic” polymers/materials with related options, for conceptually novel multi-responsive and functional materials. Therefore, we are convinced that the present findings warrant to be considered in *Nature Communications*.

To avoid controversial analogies between Pavlovian and our gel systems, we have changed the figure caption into:

Figure 2 | Logic circuits of the programmable responses in the present material system. The logic chart allows to search other materials with different stimuli and responses that show learning-inspired behaviour.

Reviewer remark:

The literature selection lacks a few key papers: For instance: (a) “Associative Learning in biochemical networks” *J. Theor. Biol.* 2007, 249, 58. (b) “A bioinspired associative memory system based on enzymatic cascades” *Chem. Commun.*, 2013, 49, 6962.

Our response:

We thank the reviewer for the recommendation of two relevant papers that we were not aware of. Their implications have been discussed above in great detail. The two papers have been added to the list of references accordingly (references 18 and 19). Note, importantly, that the articles are based on biochemical concepts, unlike the present manuscript.

Reviewer remark:

In summary, this is a solid manuscript with very good data and no objections to the results obtained. However, the interpretation is an oversimplification of the processes of learning and it would set an inappropriate example if it was published under the conceptual placement of associative learning.

Our response:

We thank the reviewer for the positive assessment regarding the data quality, but would again argue that the simplification is needed when translating natural concept into bioinspired artificial systems due to the overwhelming complexity of the natural systems. When done properly, even simple artificial system could prove to be a valuable tool and may provide inspirations for further research. On the other hand, we agree that the subtle differences resulting from simplification have to be made clear to the readers in order to avoid confusions, which we strive to achieve in the revised manuscript.

Reviewer #2:**Reviewer remark:**

In this article, O. Ikkala, A. Priimagi, and co-workers report a 'Pavlovian gel', that can 'learn' to respond to a neutral stimulus after conditioning. The agarose gel, that contains a photoacid and gold nanoparticles (AuNP) modified with lipoic acid, could be melted by an unconditioned stimulus (heat), however when light (635 and 455 nm) was irradiated into the sample nothing happened (neutral stimulus). The material could then learn to respond to light, when both heat and light were applied to the gel, the photo acid was activated, releasing protons to the medium; the AuNPs then aggregate linearly, creating a plasmonic band at 635 nm. This conditioning renders the gel responsive to light, making it possible to melt it by shining a 635 nm laser. The gel is then also coupled to out-of-equilibrium chemical reactions, making it possible for the gel to 'forget' the conditioning (urease urea chemical cycle) and the 'extinction then recovery' of the learning (hydrolysis of methyl formate).

The paper is very well written and easy to follow, and the experiments performed support the authors' claims. The ideas are very original, and will be beneficial to the field and can inspire soft matter / supramolecular chemists to pursue more life-like materials. My only criticism is that the out-of-equilibrium experiments are carried out in the melted form, which is needed for the AuNPs and chemicals diffusion, but it is not practical. The authors should more clearly indicate the limitations of their current approach. With that being said, I think that this manuscript fits well into the journal and should be accepted.

Our response:

We thank the reviewer for the positive assessment and suggestion for the revision.

We have added discussion on the limitation of the out-of-equilibrium experiments with the following text:

Since the gelling point of the 0.3% agarose gel is below room temperature, the gel just has to be left at room temperature after conditioning to allow the diffusion to take place.

Text has been added in page 11:

To be noted is that the forgetting process only works once after preparation of the gel within a short time window (~10 min), in contrast to the flexible and rewritable memory in real biological systems.

Reviewer #3:

Our general response:

We are grateful to the reviewer for the thorough review and useful suggestions to improve the quality. We have substantially revised the manuscript accordingly, especially regarding the analogy to classical conditioning, focus on in- and out-of-equilibrium processes, and distinction to conventional materials. See the revisions throughout the text, highlighted by yellow colour.

Reviewer remark:

The authors of the manuscript under review demonstrate a hydrogel system that upon simultaneous or successive exposure to heating and optical irradiation (through a range of different kinetic pathways), the dependence of gel responses, in this case melting, to optical stimuli can be tailored to mimic purely thermal cues, be reset and be recovered. The mechanism behind this altered gel behavior is reported to be the pH-dependent self-assembly (or disassembly) of lipoic acid-modified gold nanoparticles (AuNPs) in the liquid state and the resulting new plasmonic band around 635nm. The authors establish an analogy between classical Pavlovian conditioning, extinction, recovery and pathway-dependent modified, erased, reemerging gel responses, respectively, to a carefully tuned optical stimulus. To this end, the authors dub their system “Pavlovian hydrogel.” The motivation and experimental results in the manuscript are verbally and visually very clearly presented.

However, it is my strong opinion that the analogy between Pavlovian conditioning and the modified gel responses to specific kinetic pathways, as concisely illustrated by a logic circuit in Fig. 2, is controversial. Thus, the main motivation and moral of the study are not well posed, as I will summarize below. The most concrete manifestation of that is lack of some very essential control experiments, to be listed below. That said, I cannot recommend the publication of manuscript as is or with minor revisions. My suggestion is to tone the analogy down significantly and present the experimental system as a proof of concept that combines a careful design of in- and out-of-equilibrium processes in the realm of systems chemistry, while clearly distinguishing it from the existing literature of shape memory materials. This seems a daunting task, however if established, the resulting quality of the manuscript may be eligible for a second round of review.

Our response:

We thank the reviewer for the suggestions for revision and the note for distinguishing our system from shape memory materials. The phrase “memory” in our hydrogel system can easily, to our experience, lead to confusion with the shape memory materials.

[redacted]

Distinct from the above, in our case, upon conditioning-mimic programming with Stimulus 1 (unconditioned stimulus) and Stimulus 2 (neutral stimulus), the system "learns" to respond to Stimulus 2.

Text has been added in page 9:

Importantly, the gel mimicking associative learning can be clearly distinguished from conventional shape memory materials^{31,32}. This can be illustrated using a two-state classic shape memory material. Therein, the first stimulus prepares the system in a temporary shape (a kinetically trapped state), and the second stimulus recovers the equilibrium permanent shape fixed by the fabrication process. The shape memory materials also possess a memory (the equilibrium permanent shape), but the memory itself does not change after fabrication and the material always shows the same response by returning to the permanent shape. Shape memory materials therefore do not really evolve or "learn" to respond to a new stimulus, while the present hydrogel can be programmed to respond to an initially neutral stimulus.

Herein, in the new version, another focus is laid on the system chemistry approach and the difference to shape memory materials. Furthermore, we have modified the manuscript so that potentially controversial too direct analogies (associative learning, conditioning, Pavlovian gel etc.) are all toned down, and sooner the algorithmic "bioinspiration" is emphasized.

See the revised abstract:

Living systems have inspired research on nonbiological dynamic materials and systems chemistry to mimic specific complex biological functions. Upon pursuing ever more complex life-inspired nonbiological systems, mimicking even the most elementary aspects of learning is a grand challenge. We demonstrate a programmable hydrogel-based model system, whose behavior is inspired by associative learning, i.e., conditioning, which is among the simplest forms of learning. Algorithmically, associative learning minimally requires responsivity to two different stimuli and a memory element. Herein, nanoparticles form the memory element, where a photoacid-driven pH-change leads to their chain-like assembly with a modified spectral behavior. On associating selected light irradiation with heating, the gel starts to melt upon the irradiation, originally not leading to melting. A logic diagram describes such an evolution of the material response. Coupled chemical reactions drive the system out-of-equilibrium, allowing forgetting and recovery. The findings encourage to search nonbiological materials towards associative and dynamic properties.

The last chapter in the introduction (page 3):

Hydrogels have been shown to be relevant model systems for out-of-equilibrium self-assemblies and systems chemistry^{5,9}. Here we introduce a programmable soft hydrogel-based model system, the responses of which are inspired by classical conditioning, even if being mechanistically different. The aimed response is the gel is melting (gel-sol transition), which can be triggered naturally by heating where we denote this stimulus as the unconditioned stimulus, adapting the denotation from the Pavlov's dog experiment²⁰. Light irradiation with a selected spectral composition is an originally neutral stimulus, not leading to gel melting, i.e., response. The system is designed such that associating such an irradiation (the neutral stimulus) and heating (the unconditioned stimulus) triggers a memory, whereafter a subsequent sole irradiation with the given spectral composition allows the gel melting (algorithmically inspired by the conditioning process in dogs). The memory is constructed by gold nanoparticles that are initially individually dispersed in the hydrogel, which become assembled into linear chains upon stimuli association, which in turn modifies their optical absorbance. The memory of the gel can be time-wise programmed using a systems chemistry approach, which drives the hydrogel out-of-equilibrium and mimics "forgetting". Out-of-equilibrium conditions also allow spontaneous recovery of the memory after extinction.

Major remarks:

Reviewer major remark 1:

Gel programming with classical conditioning: In Pavlov's research on dog physiology, food as an unconditional stimulus and bell as a neutral stimulus are independent, or say the two stimuli "orthogonal" to each other. In the gel system however, heat and light are not orthogonal: (1) In Fig. 1C rightmost panel, starting the illumination of gel at a higher temperature, such as 29-30°C could give rise to gel melting without an emergence of a new plasmonic band. By sharp contrast, a bell would never cause salivation without prior conditioning. (2) Naturally, as reported in the Materials and Methods, the gel solution was prepared much above the melting point (at 70°C) and then stored in fridge (4 °C) overnight for gelation. What if the sample was irradiated before gelation? It appears that the initial temperature $T=24$ °C in experiments is also a fitting parameter to substantiate the analogy. (1) and (2) are missing control experiments with somewhat clear outcomes that would not serve the analogy.

Our response:

We first rationalized that irradiation and direct heating could allow at least some level orthogonality, as their effects on materials can be tuned to be different. The reviewer has correctly pointed out that in the present realization, the two stimuli are not perfectly orthogonal, i.e. the system is not ideal. However, their responses can be optimized so that we can indeed achieve the behaviour that obeys the logic diagram (Figure 2), with the apparent behaviour following the classical conditioning = associative learning algorithm.

The environmental temperature is indeed a fitting parameter, together with the laser intensity. However, if the gel is positioned in a warmer environment, such as 28°C, the laser intensity can be easily adjusted (reduced) so that the photothermal effect before conditioning is small enough as not to induce the melting, but becomes strong enough to induce melting after conditioning. The important point here is that the photothermal effect after conditioning is enhanced by more than 100% compared to before conditioning due to the change in the absorbance at 635 nm. This absorbance change is the essential feature (memory), around which other parameters, such as environmental temperature or laser intensity, can be tuned so that the system behaves in the classical conditioning way. We admit that the present system is sensitive to the operating conditions and different parameters have to be finely tuned. But as the proof-of-concept demonstration, it could still be argued that the programmed response follows the conditioning logic, even though under specific conditions. There is clearly room for optimization, but we can demonstrate already now the effect conceptually.

We have added a paragraph to explain our non-ideal system (page 7):

That the gels can be constructed to obey the logic diagram shown in Fig. 2 is relevant to illustrate that nonbiological systems can become programmable to allow responsive behaviour, inspired by associative learning. It is particularly important to note that the functionality described by the logic diagram is achieved without using electric circuits or biochemical networks. However, there are still differences in comparison to the associative learning/conditioning processes in biological systems. On one hand, in the case of the classical conditioning experiment of a dog²⁰, the stimuli (food/sound) are completely orthogonal: the two stimuli are processed through independent pathways: food-to-eyes-to-neural network and bell-to-ears-to-neural network. Each pathway undergoes physiological processes in mechanistically different ways, and finally associates to the same memory (to salivate) in the brain after conditioning. In the gel system, the two stimuli (light and heat) are not ideally "orthogonal". Specifically, light produces heat in the gel via photothermal effect and thus triggers the response. The heating effect (given by 635 + 455 nm) exists intrinsically even in a non-programmed sample because of the non-zero absorption at those wavelengths. Such intrinsic heat remains as a part of contribution to light actuation in subsequent programmed sample, while the newly obtained response (absorption from self-assembly of the AuNPs after stimuli association) contributes another part of the light induced heat (Supplementary Fig. 8).

For the second control experiment, heating to 70°C and gelation in fridge are standard preparation procedures of agarose gel and should not be considered as part of the conditioning scheme. For instance, heating, mechanical deformation and cooling are the common protocols to prepare shape memory materials. If a second heating treatment is applied after the deformation but before the cooling treatment, then the material would not show the intended shape memory behavior, since no permanent shape is programmed. But the material is still a shape-memory material. In our system, the material after gelation behaves with the same logic as in classical conditioning.

Based on the arguments above, we think the reviewer provides very good points to be clarified in the future, but the absence of the proposed experiments does not invalidate the algorithmic mimicry with classical conditioning (Figure 2).

Reviewer remark:

On line 43, the correct expression should state "... the association of the two independent stimuli..."

Our response:

This has been corrected. We thank the reviewer for the careful reading.

Reviewer remark:

Similarly, what if the gel was heated from 24 °C up by a laser at another wavelength between 550-600 nm with the same or higher power density? Again, the so-called "unconditioned stimulus" would likely directly give rise to the "conditioned response" without a learning stage.

Now the authors may argue that their analogy holds within the specific range of parameters that they have carefully tuned, the response to irradiation may be perceived as binary (melting, no melting) within the specific parameter range. But the number of parameters and (in the context of information theory) the "sloppy" response of the system to each of them makes the generalness of the concept highly questionable.

Our response:

The combination of 455 nm and 635 nm irradiation is a parameter fitted for the gel with the specific composition. Adding other wavelength of irradiation would certainly give rise to premature melting of the gel. However, as we state in the conclusion part, our system is very limited compared to real biological systems in the choice of stimuli and the operating conditions. But even in biological systems, there is a physical limit to the stimuli. For instance, if the volume of the sound is too small as not being discernible from background noise or if the sound is too loud that would be deafening to the dog, then the classical conditioning would not work either. The classical conditioning works within certain limit in biological systems, so does it in our hydrogel, though of course in a much more prescribed range.

We have added paragraph to discuss about the parameter optimization about our system (page 8):

On the other hand, in our artificial system the light contains two wavelengths, as the 455 nm wavelength is mainly for the activation of the AuNP memory to allow its new response at the second wavelength at 635 nm, which is mainly responsible for the photothermal heating after conditioning. Although mechanistic deviations exist between natural and artificial systems, importantly we can optimize the system so that its apparent behaviour mimics the association/conditioning algorithm (according to Fig. 2). Such an optimization includes selecting the working temperature, irradiation intensity and spectra of the irradiation, in a relatively flexible and adjustable range. For example, if the gel is positioned in a warmer environment such as 28°C, the laser intensity can be reduced so that the photothermal effect before conditioning is small enough as not to induce the melting, but becomes strong enough to trigger response after conditioning. The optimization is justifiable, as even systems capable of associative learning using biological cascades need optimization¹⁹. The importance is that, this system shows the

possibility for artificial materials (not based on biochemical networks or electric circuits) to exhibit the associative learning (= conditioning) algorithm.

Reviewer major remark 2:

Timing of conditioning: Apparently the duration of heat is another control parameter: what if heat was given for a much shorter time (enough that the unconditioned response appears), even to raise the temperature slightly above the melting temperature T_m , where the viscosity is still high to slow down the self-assembly of AuNPs? This further breaks the binary nature of Pavlovian conditioning and in turn the analogy: beyond T_m the rate of self-assembly is still temperature-dependent. When the heat is applied for a much longer time in Fig. 3C, then the viscosity will be low enough to mediate self-assembly. This is contrary to the neutral stimulus in fact becoming inhibitory in classical Pavlovian picture.

Our response:

In Pavlov's original experiments, the food or bell was presented to the dog and the volume of saliva produced were counted in a given time (typically 15-30 s, see I. P. Pavlov, *Conditioned Reflexes*, Oxford University Press, London, 1927.) So, the stimuli have to be presented to the subject for a certain duration in order to determine the strength of conditioning (i.e. a fitting parameter for the biological system). Similarly, we have used 10 minutes as the standard time in our system to quantify the effect, since the gel response is intrinsically slower than the neurological response.

To be more specific, the heating has been presented for 10 min in the case of forward and simultaneous conditioning, as by definition it should last at least as long as the neutral stimulus. (There are indeed different types of forward conditioning, but we chose a simple case for demonstration.) For the backward conditioning, even if the duration of heat was much longer (as far as it doesn't overlap with the irradiation), the conditioning would still be less efficient compared to the forward and simultaneous conditioning, because 10 min is already more than sufficient for the temperature to reach 50 °C inside the gel and the gel melts completely (the temperature has been checked with a thermometer). Longer heating would not result in further change of the gel properties, and that was actually the reason we chose 10 min: to make sure that there is no heterogeneity inside the gel due to partial melting.

Reviewer remark:

The explanation for the inefficiency for backward conditioning provided in lines 139-141 is hard to follow: how much does the viscosity of the 0.3% g/mL agarose solution increase upon removal of heat (hard to read the data in Ref.30 as there it is not presented in a logarithmic scale)? If the viscosity does not change by an order of magnitude, one cannot really claim an inefficiency due to the slow rate of self-assembly of AuNPs in the liquid state.

Our response:

We have performed rheological experiments of the viscosity of the melted gel (agarose is liquid state), and this has been added to the SI. The viscosity changed by roughly 100% between room temperature and 50°C. According to the Stokes-Einstein equation, the diffusion constant of nanoparticles is inversely proportional to the viscosity of the liquid. Therefore, the AuNPs should diffuse as twice as fast at 50°C compared to room temperature, which makes the backward conditioning (irradiation carried out at room temperature) less efficient compared to the simultaneous and forward conditioning.

We have added the viscosity measurement to the supporting information.

Supplementary Figure 11 | Temperature-dependent viscosity of the 0.3 wt% agarose in aqueous solution.

Reviewer remark:

So, if self-assembly is inefficient, effects of continuous variables (temperature) are at play, making the mapping between a binary response to a set of continuous responses elusive.

Our response:

In our system, the algorithmic classical conditioning logic is valid and only valid within the chosen parameter space. As the reviewer points out, it is very limited and not so “binary” compared to the biological system, but from the algorithm point of view it can be said to be an association process. To make the limitation clear for the readers, we have added discussion in the manuscript, see the answers to previous comments.

Reviewer major remark 3:

Forgetting and spontaneous recovery of memory: I am not sure how the Pavlovian picture describes “forgetting,” perhaps a reference would help. It must in principle manifest itself as a relaxation towards equilibrium from a non-equilibrium or metastable state. The author’s initial design of system disagrees with this as the gel system is in equilibrium in the absence of additional chemical cues, unless self-assembled chains of AuNPs are metastable.

In any case, both forgetting and spontaneous recovery must be associated with no additional chemical cues, whereas the authors used urea and urease for forgetting, and K₃PO₄ and methyl formate for spontaneous recovery. The addition of chemical cues in fact redefines the neutral stimulus as, e.g. illumination+urea+urease. Can the authors program their system in the first place with these modified ND, and then show forgetting and spontaneous recovery? For instance, K₃PO₄ and methyl formate would likely always disallow the self-assembly of AuNPs and either initial programming or later recovery must be compromised.

Our response:

As discussed in the response to another reviewer, we do not claim that we mimic the classical conditioning “mechanistically”, but only “algorithmically” in the apparent behaviours of the gel, allowing the logic diagram Figure 2. Neurological processes have daunting complexities that we would not claim to be able to mimic even in a simplified way. Therefore, the use of chemicals to drive the system out-of-equilibrium is a justifiable adaption in the artificial system in order to achieve the “bioinspired” algorithmic behaviour, not the same mechanism.

In the case of the forgetting experiment, the as-prepared gel already contains the urea/urease before the conditioning. **No** chemicals were added to the system after the preparation. Therefore, the neutral stimulus is still the irradiation, but the memory is modified to be the combination of AuNP assemblies + urea/urease, which makes the AuNP assemblies metastable. Therefore, we would like to argue that the necessary condition for the analogy to forgetting to be true is not that, the nanoparticle assembling process itself has to be metastable, but rather that the memory system (AuNP assemblies + urea/urease) has to be metastable. This is the case in the forgetting experiments.

On the other hand, the K_3PO_4 and methyl formate indeed were added post-synthetically. This deviates from the biological example, as we also clarify in the text that the real “spontaneous recovery” takes place without a third stimulus. This is essentially a chemical approach using clock reactions to program the life time of assembly/disassembly of the AuNPs that may be useful for out-of-equilibrium systems even without the analogy to “spontaneous recovery”. Note that in some biological networks constructed to show associative learning/conditioning (*Chem. Commun.*, 2013, 49, 6962), enzymes have been added to the system as "signals". In that sense our approach is not conceptually different. We have now clearly marked the changes.

We have added the following text to the manuscript (page 11):

In this case, the memory of the gel is in fact modified to be the AuNP/agarose pair together with the urea/urease pair, where the latter enables metastable assembly of the AuNPs.

and to page 13:

Essentially, these are all chemical approaches inspired by the biological phenomena using clock reactions to program the life-time of AuNP assemblies, which may be useful for out-of-equilibrium systems.

Reviewer major remark 4:

Materials and Methods: Absorbance measurements and methods are not described, this must be added.

Our response:

We apologize for this mistake, and the details have been added to the supporting information:

The UV-Vis spectra were measured on an Agilent Cary 5000 UV-Vis spectrometer. As reference, pure agarose gels were used, either in the gel or the sol state depending on the state of the sample.

Minor remarks:

Reviewer minor remark 1:

Lines 58-59: How fast is “fast?” What two time scales do the authors compare?

Our response:

We have optimized the composition of the gel and temperature of conditioning, so that the change of absorbance (assembly of AuNPs) is comparatively fast without the need for excessive heating (heating to 70°C for instance, which results in fast degradation of the photoacid. Here we select the “fast” association process after screening the temperatures, gel compositions, and AuNP sizes (Fig. S2-S6). In the end, 13 nm AuNPs and 0.3wt% agarose proves to provide the most desirable response. The time scale for conditioning was also chosen to be between 10 min and 60 min depending on the situation.

Reviewer minor remark 2:

Line 170: Did the authors mean “540nm?”

Our response:

We actually meant 635nm. As the sentence may cause confusion, we have rewritten it for clarity as following (page 11):

The decrease of the absorbance at 635 nm is significant, indicating that the AuNPs are well protected by the ligands during the self-assembly process so that they can also be disassembled.

Reviewer minor remark 3:

Fig. 4E: The green and blue curves are very hard to distinguish in print.

Our response:

The light blue curve has been replaced by dark blue in Fig. 4E for clarity.

REVIEWERS' COMMENTS:

Reviewer #1 (Remarks to the Author):

The article has been greatly improved by incorporating larger clarifications and also corrections of misconceptions.

The authors make an important point that bioinspired design is about abstracting principles and reformulating them into synthetic material concepts, but still one needs to be aware to what extend bioinspired design is still upheld or whether too many elements have been removed to still claim the translation of the principle. This is a very critical aspect to promote bioinspired material concepts. After removing the unfounded claims of Pavlovian adaptation, the manuscript feels much more in balance between claims of interweaving signals and most simplistic effects of learning. One can definitely argue that the increased discussion in the manuscript provides a contribution to the discussion of "what could be a learning material", and as such it is relevant and it is also relevant to Nature Communications.

In a down to earth assessment, the manuscript mostly shows the installation of a metastable memory that can be activated with a different signal following two orthogonal signal processing pathways. It follows the logic gating of the form IF ...[X] AND [Y] ... THEN. Yet, in terms of semantics, one could argue that this potentially shows a very primitive feature of associative learning. Moreover, installation of a memory, even though it is external programming of a memory rather than internal learning by concatenation of signal processing pathways, is in general an important aspect in adaptive materials.

I have one more comment on the logic diagram. The "OR" module in the memory does not make sense. It does not make much sense to use the output of an "OR" logic gate as an input for exactly the same gate. This is like stating (blue AND yellow) OR Grey  Grey. This should be corrected.

Reviewer #2 (Remarks to the Author):

The revised version is ok for me.

Reviewer #3 (Remarks to the Author):

I believe the authors have largely addressed my concerns. However, after detailed revisions, I believe the manuscript is still very much immersed in the concepts of associative learning and the Pavlovian picture, although the authors claim that these concepts only provide an inspiration to the design of the material at hand and its response pathways. Inspiration dictates that the contrasts between their system and classical conditioning & associative learning should be brought up much earlier in the manuscript. Before formal acceptance, the authors should revise the first three pages accordingly. The non-orthogonality of the signals is still an issue; beyond being merely ideal or not. Thus, it needs to be highlighted properly earlier on in the manuscript.

One more important note: In response to the Reviewer 1, the authors have written "Related to this article, we do see a conceptual difference. In Chem. Comm., 2013, 49, 6962 the chemical composition is modified by adding new enzymes (as "signals") to allow a new response. By contrast, in our case, our chemical composition remains fixed, and the stimuli are physical external conditions. We found it important to keep the chemical composition fixed in the learning process, in analogy with the Pavlov's classical conditioning experiment where the state of the dog is kept as constant as possible." However, for spontaneous recovery, the authors have stated later in their rebuttal that "K3PO4 and methyl formate indeed were added post-synthetically." The conceptual similarities and differences to the article suggested by the Reviewer 1 must be discussed in the manuscript further based on the applications for forgetting and spontaneous recovery.

I recommend conditional acceptance provided that the above points are addressed, to make the reader aware in the introduction to the paper of the shortcomings of the analogy between the hydrogel system and associative learning concepts.

Answers to the Reviewers' comments:

Reviewer #1

Reviewer remark:

The article has been greatly improved by incorporating larger clarifications and also corrections of misconceptions.

The authors make an important point that bioinspired design is about abstracting principles and reformulating them into synthetic material concepts, but still one needs to be aware to what extent bioinspired design is still upheld or whether too many elements have been removed to still claim the translation of the principle. This is a very critical aspect to promote bioinspired material concepts. After removing the unfounded claims of Pavlovian adaptation, the manuscript feels much more in balance between claims of interweaving signals and most simplistic effects of learning. One can definitely argue that the increased discussion in the manuscript provides a contribution to the discussion of “what could be a learning material”, and as such it is relevant and it is also relevant to Nature Communications.

In a down to earth assessment, the manuscript mostly shows the installation of a metastable memory that can be activated with a different signal following two orthogonal signal processing pathways. It follows the logic gating of the form IF ...[X] AND [Y] ... THEN. Yet, in terms of semantics, one could argue that this potentially shows a very primitive feature of associative learning. Moreover, installation of a memory, even though it is external programming of a memory rather than internal learning by concatenation of signal processing pathways, is in general an important aspect in adaptive materials.

Our response:

We thank the Reviewer for the efforts. We appreciate the reviewer's assessment that the revised manuscript demonstrate “a generally important aspect in adaptive materials”, and is “relevant to Nature Communications”.

Reviewer remark:

I have one more comment on the logic diagram. The “OR” module in the memory does not make sense. It does not make much sense to use the output of an “OR” logic gate as an input for exactly the same gate. This is like stating (blue AND yellow) OR Grey  Grey. This should be corrected.

Our response:

We thank for the comments. The OR gate here is a pure logical operator to represent a sustained memory once it has been switched on, without further input of the learning AND gate. This is important for the memory to remain activated after the two stimuli (light + heat) have been removed, since the “switched-on” memory can be activated itself by the OR gate. Without the OR gate, both heat and light have to be present to activate the output of the memory. This would mean that the system cannot remember the conditioning. Therefore, we believe that the OR gate should be kept in the diagram. A comparable logic gate design based on genetic circuits has been adopted by another research group, see Fig. 1c in *Nat. Commun.* **5**, 3102 (2014), indicated as Ref 16 in our manuscript.

On the other hand, we agree that this does not mean that the material has a real OR gate. To make this point clear, we have added the following sentences in the caption of Figure 2:

Note that the OR gate is a pure logical operator to represent a sustained memory once it has been switched on, without the need of further input from the learning AND gate. The memory of the hydrogel does not possess a real OR gate.

Reviewer #2**Reviewer remark:**

The revised version is ok for me.

Our response:

We thank for the reviewer's efforts.

Reviewer #3**Reviewer remark:**

I believe the authors have largely addressed my concerns. However, after detailed revisions, I believe the manuscript is still very much immersed in the concepts of associative learning and the Pavlovian picture, although the authors claim that these concepts only provide an inspiration to the design of the material at hand and its response pathways. Inspiration dictates that the contrasts between their system and classical conditioning & associative learning should be brought up much earlier in the manuscript. Before formal acceptance, the authors should revise the first three pages accordingly. The non-orthogonality of the signals is still an issue; beyond being merely ideal or not. Thus, it needs to be highlighted properly earlier on in the manuscript.

Our response:

We thank for these comments. We have carefully re-considered the introduction section (first three pages) and brought out the difference between natural and our artificial system in a much earlier place. The issues about non-orthogonality and independency in stimuli-response pathway are discussed in the introduction. Please see the added text in the Introduction section as shown below.

In this case, the stimuli are fully orthogonal and undergo independent pathways (visual or auditory sense), finally being associated in the brain that provides the memory. For artificial materials, it is very challenging to construct two entirely orthogonal pathways for a specific stimuli-response, due to the lack of a central neural network. However, in a reductionistic approach, a synthetic material may algorithmically show some aspects of associative learning, provided that the system is capable of responding to two different stimuli, combined with a built-in memory element.

and

Although the two stimuli are not fully orthogonal and the response pathways (heating and light-induced heating) are interdependent, which deviates from the ideal concept, the programmed hydrogel system algorithmically shows many aspects of biological learning, and thus encourages research in synthetic materials towards more associative and dynamic functionalities.

Reviewer remark:

One more important note: In response to the Reviewer 1, the authors have written "Related to this article, we do see a conceptual difference. In Chem. Comm., 2013, 49, 6962 the chemical composition is modified by adding new enzymes (as "signals") to allow a new response. By contrast, in our case, our chemical composition remains fixed, and the stimuli are physical external conditions. We found it important to keep the chemical composition fixed in the learning process, in analogy with the Pavlov's

classical conditioning experiment where the state of the dog is kept as constant as possible.” However, for spontaneous recovery, the authors have stated later in their rebuttal that “K3PO4 and methyl formate indeed were added post-synthetically.” The conceptual similarities and differences to the article suggested by the Reviewer 1 must be discussed in the manuscript further based on the applications for forgetting and spontaneous recovery.

Our response:

We thank for these comments. The reviewer is correct that for the extinction and spontaneous recovery experiment (Fig. 4 d-f), additional chemical cue is added into the programmed gel. The aim of this experiment is to explore the possibility of extinguishing memory by using external stimulus, and it is mechanistically different from biological systems, which is well described in the text. On the other hand, the other programming processes including association (heat/light induced self-assembly, Fig.1), timing dependency (Fig.3), and forgetting (Fig. 4, a-c) are ALL achieved under the condition where the material composition remains unchanged.

Regarding Review’s concern, we have added further discussion about the similarities and differences between the system demonstrated by the referenced article (suggested by Review 1) and the one in this manuscript. A paragraph has been added to the “Discussion” section, highlighted below.

Associative responsive properties have been exploited in systems with reversible polymerization¹⁸ and enzymatic reaction reactions¹⁹. Very often, a modification of chemical composition is needed during the association process, for example by adding enzyme into reaction¹⁹, similar to the extinction and spontaneous recovery experiments in our programmed hydrogel (Fig. 4 d-f). Although these approaches provide valuable examples of artificial systems inspired by biological processes, we believe that keeping the chemical composition unchanged in the association process (Fig. 1, Fig. 3 and Fig. 4 a-c) will bring the analogy closer towards Pavlov’s classical conditioning experiments. Besides, such chemically fixed systems allow realizations of future devices with dynamic adaptation – once fabricated, the functions can be switched on via associative learning and recovered by forgetting – without the addition of any post-synthetic chemical fuels. Further clarifications of similarities and differences between programmed hydrogel and the reported enzyme-based systems are not enumerated here due to the complexity of each systems. We refer readers to Peer Review File in Supplementary Information for more details.

Regarding the conceptual similarities and differences to the articles suggested by Reviewer 1, in the response letter in the first round of review, we have made very detailed explanation and discussion composing of around 1200 words. We believe that with the added paragraph at the end of the manuscript, together with the Peer Review File that will be published online, difference between those material systems should be now clearly visible for the readers.

Reviewer remark:

I recommend conditional acceptance provided that the above points are addressed, to make the reader aware in the introduction to the paper of the shortcomings of the analogy between the hydrogel system and associative learning concepts.

Our response:

We have revised the manuscript to address all the points raised by Reviewer. We have modified the introduction to bring out the difference between natural and artificial systems, the shortcomings of the analogy, non-orthogonality, and comparison to the article suggested by Review 1. We believe the bio-inspired concept and its limitations are now clearly stated to the readers.